# Strategies to Meet Nutritional Requirements and Reduce Boar Taint in Meat from Entire Male Pigs and Immunocastrates

**DOI:** 10.3390/ani10111950

**Published:** 2020-10-23

**Authors:** Giuseppe Bee, Nathalie Quiniou, Hanne Maribo, Galia Zamaratskaia, Peadar G. Lawlor

**Affiliations:** 1Agroscope, Institute for Livestock Sciences, La Tioleyre 4, 1725 Posieux, Switzerland; 2IFIP-Institut du Porc, La Motte au Vicomte, 35650 Le Rheu, France; nathalie.quiniou@ifip.asso.fr; 3Pig Research Centre, Danish Agriculture & Food Council, Axeltorv 3, DK-1609 Copenhagen, Denmark; HMA@seges.dk; 4Department of Molecular Sciences, Swedish University of Agricultural Sciences, Box 7015, 750 07 Uppsala, Sweden; Galia.zamaratskaia@slu.se; 5Pig Development Department, Animal and Grassland Research and Innovation Centre, Teagasc, Moorepark, Fermoy, P61 C996 Co. Cork, Ireland; peadar.lawlor@teagasc.ie

**Keywords:** energy, protein, nutrition, pork quality, management

## Abstract

**Simple Summary:**

Raising entire male pigs and immunocastrates poses various challenges on how the nutrient requirements of these growth-efficient animals can be met. Applying the appropriate nutritional strategy will markedly affect the production efficiency and the carcass and pork quality, but will also control the incidence of boar-tainted carcasses.

**Abstract:**

This paper reviews the current knowledge on the nutritional requirements of entire male and immunocastrated pigs to obtain an efficient growth, low boar taint level, and good carcass and meat quality. We present the reasons for offering entire males ad libitum access to the diets in order to optimize their protein deposition potential. Boar taint is one of the major issues in the production of entire males; therefore, the impact of various skatole- and indole-reducing feed ingredients is discussed regarding their efficiency and the possible mechanism affecting skatole and indole production in the hindgut. Entire males have lean carcasses, so their intramuscular fat content can be lower than that of surgical castrates or females and the adipose tissue can be highly unsaturated. The possible nutritional strategies to counteract these effects are summarized. We conclude that immunocastrates can be fed similarly to entire males until the second vaccination. However, due to the metabolic changes occurring shortly after the second vaccination, the requirements for essential amino acids are markedly lower in immunocastrates than in entire males.

## 1. Introduction

Increasing concerns regarding animal welfare have now placed the surgical castration of male pigs at an early age without pain relief under great scrutiny in major European pig-producing countries [1]. Various solutions, such as castration under anesthesia and analgesia, raising entire male pigs, or immunization against gonadotrophin (also referred to as immunological castration or immunocastration), are available and make sense from a point of view of production efficiency. However, some pig-meat importing countries will not accept meat from entire males or from immunocastrates, which poses a problem for countries with large export markets. Compared to surgically castrated pigs, entire male pigs are more nutrient efficient as their empty bodies and carcasses contain more protein and less fat [2,3,4]. However, the carcasses of entire males have greater portions of fore-end and lower portions of ham, their muscles contain less protein, and the meat is less tender [5]. Because of this, and despite the advantages with respect to production efficiency, it is still common practice to castrate male pigs. The principal reason for this is to reduce the possible risk of the offensive odor and flavor known as boar taint that is primarily caused by an increase in the concentration of the compounds androstenone, skatole, and indole in the meat. Boar taint develops to various degrees in sexually mature as well as immature entire males and greatly influences the consumer’s acceptability of pork [6]. 

An already proven alternative to early surgical castration of male pigs is immunization against gonadotrophin-releasing factor, or immunocastration, and this greatly reduces the incidence of boar taint in male pigs [7]. Immunocastration suppresses testicular development and function, so these formerly male pigs become immunological castrates, thereby progressively reaching a productivity level similar to that of surgical castrates while lacking the expression of the offensive odor and flavor. A further advantage of immunological castration is that these male pigs are allowed to grow as entire males for most of the growing–finishing period, thereby benefiting from the naturally greater feed efficiency and improved carcass composition of entire males [8,9,10] during this time, as the immunization is normally conducted in the late finishing stage. 

From a nutritional point of view, raising male pigs without surgical castration poses three major challenges. The first challenge is to adapt the composition of the diet to cover the nutritional needs for an efficient protein deposition without excessive feed cost and environmental impact due to nutrient spillage. The second is to select the dietary ingredients that reduce skatole and indole production so that bacterial degradation of tryptophan in the large intestine is minimized, thereby lowering skatole and indole production and their expression in the tissue. The third is to optimize the diet composition for immunocastrates following the second vaccination to take into account the change in metabolic demands of pigs, which, at this time, switch from being those of entire males to demands more akin to those of castrates. The objective of this short review is to share current knowledge gathered during the European Cooperation in Science and Technology (COST) Action entitled “Innovative Approaches for Pork Production with Entire Males” (IPEMA) on the main challenges with respect to the nutrition of entire males and immunocastrates. 

## 2. Assessment of Energy Requirements

The factorial approach to determine nutritional requirements identifies three energy sinks that determine the differences in energy requirements between females, surgical castrates, and entire males (and immunocastrates): first are the requirements for maintenance, second are the requirements for muscle (or protein) growth, and third are the requirements for fat (or lipid) deposition. Furthermore, a portion of ingested dietary energy is not available for maintenance and growth but is instead used for physical activity. In entire males, this can be quite substantial due to the relative increase in social interactions among penmates, such as fighting, biting, and mounting. As described in the literature, protein deposition increases linearly with energy intake up to a maximum value (called a plateau [Figure 1]) and this depends on the amino acids available or the growth potential of the animal [11,12]. Above this value, protein deposition does not increase further, but lipid deposition does increase, as does the resulting lipid-to-protein ratio in the marginal body weight gain. Therefore, the energy intake at which protein deposition reaches the plateau (also known as the protein deposition potential), when not limited by amino acid supplies, is considered the optimal energy allowance for a pig and corresponds to the optimum average daily gain and optimum feed conversion ratio. Above this level, the extra energy intake is associated with a poor marginal increase in average daily gain but a high marginal decrease in feed efficiency due to the high marginal increase in fat deposition. However, after decades of genetic selection for lean tissue growth, no plateau is observed in the modern entire male genotype [2]. Therefore, as long as the amino acid requirements of the pig are met, protein deposition is an energy-dependent process during the growing-finishing period. Therefore, to exploit the full potential of protein deposition, entire males should have ad libitum access to their diet [13]. 

The current literature shows no difference in the lean meat content in the carcasses of modern types of male pigs associated with a different level of nutrient intake under different feeding conditions, i.e., when male pigs are provided with ad libitum access to dry feed vs. restricted access to dry feed in France [14] or to wet feed in Denmark [15]. The corollary of this conclusion is that any factor that affects spontaneous feed intake affects both protein and lipid deposition, and consequently muscle gain and growth rate, rather than mainly fat deposition as in the case of surgical castrates or the fat type of pigs. In immunocastrates, the increase in energy intake observed after the second immunization is associated with an increased lipid proportion in body weight gain [16,17,18] that corresponds to a situation where energy supply is no longer limiting for protein deposition.

## 3. Energy Concentration

Due to the limited appetite of entire males, which is caused by both androgens and estrogens [19] and the less time spent at the feeder [20], a decrease in energy intake utilized for growth limits the overall protein deposition during the growing-finishing period. This means that any decrease in energy intake is associated with a reduction in both protein and lipid gains. On the contrary, when the average daily weight gain is far below the expected growth potential of the pig, even when provided with ad libitum access to feed, increasing the dietary energy concentration is seen as a solution to overcome the limitation of the low ingestion capacity in the pig [21]. This strategy is efficient in female and castrated pigs [22], but the effects of increasing dietary energy density are more complex to analyze in entire males. Feeding entire males with 7% extra energy and 14% extra protein resulted in a 6% increase in daily weight gain, a 5% increase in feed utilization, and 4.6 days shorter time to reach the slaughter weight. 

When formulating diets for entire males, the productivity and the production of skatole in the intestine should be considered. Increased energy digestibility in the diet means less indigestible energy available for the microbes in the large intestine to produce boar taint components [23], while also orientating the selection of a microbiota more prone to use the indigestible tryptophan as an energy fuel instead of microbial protein. A net energy (NE) concentration in the diet between 9.5 and 10.0 MJ/kg allows flexibility to formulate diets for entire males. However, when the NE level of the diet is above 10.0 MJ/kg, it is more difficult to design diets with a minimum level of indigestible compounds, like soluble or fermentable fibers, in an effort to lessen the production of skatole in the hind gut.

## 4. Feeding Level

In some European countries, finisher pigs are restrictively fed in order to improve carcass leanness and feed efficiency. This is due to the great appetite of surgical castrates approaching slaughter weight and their propensity to deposit excess dietary energy as adipose tissue. Feed restriction is an efficient strategy to improve feed efficiency when the decrease in energy supply limits lipid deposition rate without negatively affecting protein deposition.

With liquid feeding, the use of long troughs allows the practice of restricting feed allowance as would be required for surgical castrates (or, in some instances, females) at heavy target slaughter weights. Practicing ad libitum liquid feeding where pigs are fed from long troughs is much more difficult to implement, due to increased feed wastage. However, newer ad libitum, sensor-controlled short-trough liquid feeding systems allow more frequent feeding and ad libitum feeding of entire males works well with these systems. Competition among pigs occurs only during the delivery of the meals with liquid feeding; however, negative social interactions can occur when pigs are fed below their appetites [20,24,25]. Due to the adverse impact of restriction on behavior and to improve weight gain for some breeds, ad libitum feeding is recommended for entire males in the growing-finishing phase. Nevertheless, ad libitum sensor-controlled liquid feeding can also create a scramble for pigs to feed, even when there is existing feed in the trough, and this is thought be triggered by the noise from the feeding valve opening at the time of feed-out.

## 5. Amino Acid Requirements

There is general agreement that, in relation to the composition of the body weight gain, the requirement for essential amino acids per unit of energy is greater in entire males than in surgical castrates and female pigs. Quiniou et al. [26] estimated that the standardized ileal digestible (SID) lysine requirement is 0.1 g/MJ NE greater (grower period: 0.94 vs. 0.81 g/MJ NE; finisher period: 0.81 vs. 0.71 g/MJ NE) for entire males than for surgical castrates throughout the growing-finishing phase (Figure 2). Aymerich et al. [27] found a 17% greater SID lysine requirement for entire males (0.91 vs. 0.75 g SID/MJ NE) than for female pigs, especially in the finisher period. Interestingly, Moore et al. [28] hypothesized that entire males might use the available amino acid more efficiently for growth compared to female pigs. The body weights were 50 and 100 kg when female pigs and entire males were offered the same diet (0.5 g standardized ileal digestible lysine/MJ digestible energy) that supported near maximum growth rate. Both sexes had similar lysine intake but entire males grew 13.2% faster and were 10.4% more feed efficient than females (Figure 3). Maribo et al. [29] showed that entire males could effectively benefit from 17–25% more amino acids (lysine, methionine, threonine, and tryptophan) than recommended by the Danish standards for better growth and feed utilization.

Increasing environmental concerns in recent years have led to improvements in the feed efficiency as well as in the practice of formulating pig diets with reduced protein and essential amino acid content to reduce nitrogen losses and ammonia emissions. This practice is feasible in surgical castrates and female pigs, but not in entire males, as shown in a recent study by Ruiz-Ascacibar et al. [2]. The authors compared the protein and fat deposition rates of surgical castrates and entire males and showed that entire males had a significantly lower protein deposition rate when offered an amino acid- and protein-restricted diet (20% lower than the standard) (Figure 4). The authors concluded that, unlike with surgical castrates, reducing the essential amino acid supply for entire males below the current recommended levels (which are higher than those of females and surgical castrates) has negative consequences for pig growth. Furthermore, this negative effect will worsen if the essential amino acid profile of the diet is unbalanced. However, when the undersupply of essential amino acids is limited to a short period of restriction (for instance, during the grower period), entire males with high lean-tissue growth potential can express compensatory protein deposition during the re-alimentation period (finisher period), and, consequentially, achieve whole body protein deposition targets [30].

## 6. The Special Requirements of Immunocastrates

As previously mentioned, prior to the second vaccination (booster) against gonadotrophin-releasing factor, which is usually administered four to six weeks prior to slaughter, entire males and immunocastrates are similar in terms of growth performance and composition of body weight (BW) gain. Therefore, to maximize growth and obtain maximal protein deposition rates, the same recommendations for nutrient requirements can be used for both. Within two weeks after the booster injection, the testosterone levels drop in response to the lack of luteinizing hormone [7,31]. Subsequently, the metabolism of immunocastrates progressively switches toward that of surgical castrates, resulting in increased appetite [13,32], reduced growth, and poorer feed utilization (personal communication, Maribo 2020). While more fat is retained in immunocastrates after the second immunization, particularly due to their increased feed intake, a reduced protein-to-lipid ratio also becomes apparent in the BW gain. For this reason, the immunocastrates require a lower essential amino acid-to-energy ratio in their diet. The lysine requirement of immunocastrates decreases by up to 30% compared to that of entire males at approximately 14 d after the second vaccination (Figure 5; [13,16,33]). With immunocastrates, when switching to a final finisher diet after the booster is not feasible, a possible alternative would be to restrict feed intake, thereby improving feed efficiency. This would then reduce the SID lysine and protein intake (as well as energy intake) and as a positive side effect nitrogen emission. However, quantitative feed restriction after the second immunization is associated with impaired growth, although feed efficiency and carcass leanness are not improved [34]. Feed restriction also results in a greater number of skin lesions in pigs, which are indicative of greater negative social interactions among penmates. After the second immunization, the duration of individual meals (min/meal) and feeding rate (g/min) both increase [35], all at a time when access to feed at a trough is restricted, thereby explaining the greater animal-to-animal interactions.

## 7. Dietary Control of Boar Taint

Dietary composition is one of the major factors regulating intestinal skatole and indole synthesis. Several mechanisms control involving dietary components can affect skatole and indole levels. First, diet can reduce the availability of the skatole and indole precursor L-tryptophan in the colon by changing the rate of apoptosis. Second, diet can alter the composition of gut microbiota and the availability of *Bacteroides* spp. and *Clostridium* spp., which have been reported to produce skatole [23]. Some dietary components can increase energy availability and shift microbial activity from proteolytic to saccharolytic. Diet also affects intestinal transit time and the rate of absorption of skatole and indole. Details of these factors are described by Wesoly and Weiler [23]. Moreover, bioactive compounds in feed can affect the skatole and indole levels in fat by induction or inhibition of their metabolism via cytochrome P450 (CYP450) enzymes [36].

A substantial effort has been made to reduce skatole levels by dietary means. Different feeding regimes and dietary compositions have been tested for their effects on skatole and indole levels in back fat, blood, intestinal digesta, and feces, and the results from these studies are well reviewed [23,37]. In the present article, we summarize and discuss the research results published since 2010. We paid particular attention to additives that both reduce the levels of boar taint compounds and promote animal health.

### 7.1. Chicory Root and Inulin

Accumulating evidence, albeit not always consistent due to differences in experimental design, supports the skatole-reducing effects of dietary supplementation with chicory root. Inclusion of purified inulin in the diet produced similar results to chicory root, suggesting that inulin, a fermentable fiber also found in large amounts in the Jerusalem artichoke, is the main chicory component responsible for the skatole reduction (Table 1). Feeding chicory at 15% for 14 days before slaughter successfully reduced skatole [38]. However, chicory is very expensive, so a shorter period was tested under Danish conditions to make it more economically suitable. Continuous feeding 15% chicory in the diet for 3–4 days before slaughter was sufficient to reduce skatole by 50–60% [38,39].

Indole levels are usually not affected by the presence of either chicory root or inulin in the diet (Table 1). Aluwé et al. [42] suggested that the addition of chicory or oligosaccharides can decrease the bacterial conversion of tryptophan to skatole while increasing its conversion to indole. Although speculative, the absence of effects on indole might be regarded as a positive finding, given that indole makes only a minor contribution to boar taint and positively affected intestinal health in other species [50,51]. One point to be emphasized is that, to the best of our knowledge, the role of indole in pig health has not been studied.

Inulin, a high-molecular weight polymer of fructose, is fermented by endogenous microbiota, with a resulting increase in the production of short-chain fatty acids (SCFA), such as acetate, propionate, and butyrate [52]. Butyrate, in particular, is beneficial because of its anti-inflammatory properties [53,54]. Lepczyński et al. [55] demonstrated that dietary supplements containing dried chicory root and inulin altered the expression of several hepatic proteins associated with energy metabolism and induced the expression of proteins involved in protection against oxidative stress. Moreover, a study using an in vitro model of the pig’s gastrointestinal tract reported an upregulation of genes related to intestinal barrier integrity following treatment with inulin [56].

The exact mechanism of the skatole-reducing effect of chicory root is not fully understood, and several explanations have been offered. One is that the addition of chicory or other sources of fermentable dietary fibers promotes an increased butyrate concentration in the colon, which in turn reduces cell apoptosis and, therefore, the availability of tryptophan for skatole synthesis [57]. Another explanation is that the compounds in chicory root modulate the microbial composition and decrease the abundance of skatole-producing bacteria. Nonetheless, neither of these two hypothesis has been confirmed [41,58]. Li et al. [41] and Øverland et al. [58] explained the reduction in skatole after feeding chicory as a response to increased microbial activity and an associated increased incorporation of amino acids into bacterial biomass, resulting in an inhibition of skatole synthesis from tryptophan. Moreover, Rasmussen et al. [59] suggested that the skatole-reducing effect of chicory might be due, at least in part, to increased activities of the skatole-metabolizing enzymes CYP1A2 and CYP2A19.

### 7.2. Hydrolysable Tannins

Another promising feed additive with potential to reduce the levels of skatole and indole in the adipose tissue is hydrolysable tannins, which are secondary metabolites found in many plants [45,46]. Tannins inhibit proliferation and apoptosis in the cecum; this inhibition then reduces intestinal skatole production due to the lower availability of cell debris and tryptophan [60]. Indeed, intestinal skatole production was lower in pigs supplemented with tannin [45]. The effects on skatole and indole levels in back fat, however, were not consistent, ranging from reduction to induction depending on the dose of tannins used (Table 1). Thus, the future use of hydrolysable tannin supplementation to reduce boar taint will first require determination of the optimal dose. Interestingly, Bee et al. [46] observed a numerical reduction in another boar taint compound, androstenone, in pigs fed hydrolysable tannins. A later study confirmed these results, as the androstenone levels in the adipose were also numerically lower in entire males fed the tannin supplemented diet [47]. However, the pigs in the latter study had low levels of androstenone overall. More studies are needed to confirm the effect of hydrolysable tannins on androstenone and to elucidate the underlying mechanism.

Generally, tannins are viewed as antinutrients, as they form complexes with proteins and minerals, as well as with digestive enzymes, and they interfere with nutrient digestibility. However, tannins seem to be relatively tolerable to pigs compared to other animals, and pigs can consume tannin-rich feedstuffs without negative health consequences [61]. Moreover, recent findings indicate that tannins also display antibacterial effects and can potentially serve as alternatives to in-feed antibiotics for farm animals [62]. Other observations indicate that tannin-rich wood extracts can improve feed efficiency in weaned piglets and reduce intestinal bacterial proteolytic reactions [63]. For example, addition of a 2% chestnut-tannin extract reduced the incidence and diarrhea severity of weaned pigs [64,65].

### 7.3. Raw Potato Starch

Previous studies have demonstrated that the addition of raw potato starch in feed consistently reduced skatole levels (reviewed by Zamaratskaia and Squires [37], Wesoly and Weiler [23]). In these studies, raw potato starch was provided to the pigs as a top dressing, which is not practical in commercial pig production. However, formulation of 20% raw potato starch into a pelleted diet did not reduce skatole levels in the distal intestine or in the adipose tissue of entire males [58].

When present, the skatole-lowering effect of potato starch is thought to be due to its high concentration of resistant starch. The effect of dietary resistant starch alters the intestinal microbial composition, increases the production of short-chain fatty acids, reduces damage to colonocytes, modulates lipid metabolism, and improves colonic mucosal integrity [48,66]. However, practical solutions for incorporating raw potato starch into the pig’s diet are required.

### 7.4. Other Ingredients

A recent study demonstrated that feeding a diet with blue sweet lupin seeds altered microbial activity in the large intestine of pigs [67]. In that study, a high level of lupin seeds in the diet (300 g/kg) tended to decrease the indole concentration in the distal colon below that seen with a low inclusion level (150 g/kg). Skatole concentrations were not measured. Inclusion of either chicory root or sweet lupin in the diet increased colonic abundance of *Bifidobacterium thermoacidophilum* and *Megasphaera elsdenii* [68].

Holinger et al. [69] attempted to reduce skatole and indole concentrations by feeding grass silage to Swiss Large White pigs. These authors hypothesized that the presence of fermentable energy (fiber-rich diet) would promote the use of tryptophan for microbial protein synthesis instead of its conversion to indole compounds in the hindgut. However, neither skatole nor indole levels were reduced in this study.

Several attempts have been made to reduce skatole and indole production by adding sugar beet pulp to the pig’s diet (reviewed by Wesoly and Weiler [23]). Lately, only a few experiments have been performed to investigate the effect of Jerusalem artichoke on boar taint. Vhile et al. [44] reported a tendency toward decreased skatole levels in the hindgut and adipose tissue, whereas indole levels were not affected. The reduction in skatole levels was explained by a decrease in *Clostridium perfringens* and an increase in short-chain fatty acid production in the intestine, which reduced the pH in the digesta. Pieper et al. [49] suggested that sugar beet pulp reduced skatole synthesis in the proximal large intestine. The addition of a mixture of lignin and cellulose into the diet also reduced skatole levels [49]. In contrast to inulin, cellulose has low solubility and is not readily used as a substrate by intestinal microbiota. The mechanism behind this reduction warrants further investigation.

Pieper et al. [49] also observed reduced skatole formation in the distal colon after feeding a lignin- and cellulose-containing diet. The authors suggested that the effect was due to combining soluble and insoluble fiber sources to provide fermentable carbohydrates to the intestinal microbiota along the entire gut. In that study, skatole formation in both the proximal and distal large intestine was reduced. Another and much cheaper way to reduce skatole is by feeding pure cereal grains for 3–4 days before slaughter, as it reduces skatole by 50% [70].

## 8. Carcass and Meat Quality Are Not Related to Boar Taint

An undisputed finding is that, when reared under identical conditions, the carcasses of entire males have a lower dressing percentage, greater lean meat content, and a more extensive longissimus thoracis muscle cross-sectional area when compared to surgical castrates [8,71]. Furthermore, Aaslyng et al. [5] showed that the conformation of carcasses of entire males differs from that of surgical castrates, as the entire males have larger fore-ends and smaller hams and their meat has a lower protein content. The lower carcass fat content is associated with a lower intramuscular fat content [8]. Depending on the average intramuscular fat content of the genetic line used, the decrease observed in entire males compared to surgical castrates or female pigs might impact the sensory traits of pork, like its texture and taste [72]. The lower fat deposition usually results in a greater degree of unsaturation of the adipose tissue [73]. From the human nutrition point of view, a greater amount of polyunsaturated fatty acids can be considered beneficial [74]. However, from a meat-processing perspective, soft tissue with an elevated level of mono- and poly-unsaturated fatty acids is less suitable [75,76,77]. The question arises as to how these “defects” could be avoided by nutritional strategies.

The fatty acid composition of pork can be easily manipulated by selecting appropriate feed ingredients. This is because, in pigs, the amount of ingested polyunsaturated fatty acid is closely related to the amount deposited in both the subcutaneous and intramuscular lipids [73,78,79]. In countries where the adipose composition is a quality trait, feeding recommendations need to be developed based on the fat composition of the diet [80]. However, the currently available recommendations were developed for surgical castrates and female pigs and, depending on the amount of deposited adipose tissue, they might not be valid. A valid strategy to increase the amount of deposited intramuscular fat without altering carcass leanness is to reduce the protein/lysine-to-energy ratio of the diet [81] below the requirement for some period prior to slaughter. However, one has to be aware that applying this feeding strategy would also reduce pig growth in that period prior to slaughter. To our knowledge, these strategies have not yet been tested in entire males but only in pigs with a potential for intramuscular fat deposition [82]. Thus, whether this feeding strategy would be successful with entire males of modern lean breeds remains to be established. With regard to the carcass and meat quality of immunocastrates, these appear to be similar to those of surgical castrates [10], which implies that no specific dietary regimen targeting meat quality traits is necessary with immunocastrates.

## 9. Practical Considerations with Feeding Entire Males or Immunocastrates

Ireland and the UK have been producing entire males since the 1970s because entire males grow faster and are more feed efficient than their surgical castrate counterparts. For instance, Hanrahan [83] reported an 8% improvement in carcass feed efficiency and a 6% increase in carcass average daily gain when entire males were compared with surgical castrates. Likewise, Lawlor et al. [84] found an 8.4% improvement in the feed conversion ratio in entire males compared with surgical castrates, although no increase was found in average daily gain. Traditionally, producers in these countries have been able to benefit from the reduced production costs associated with producing entire males since the carcass weight was low relative to that in other European pig producing countries and so the risk of boar taint was also lower. However, because it is very cost beneficial for farmers to produce heavier pigs, and because processors benefit from savings in per unit processing cost, as well as larger joints (preferred by some sectors of the processing and catering industries) with heavier carcasses, the slaughter weight has increased substantially in recent years. In 2000, the average slaughter weight in Ireland was 90.1 kg live weight (68.1 kg carcass weight) and by 2019 it had increased to 118.4 kg live weight (90.3 kg carcass weight) [85]. This 28 kg increase in slaughter weight may seem large; however, the age of pigs at slaughter has decreased by 1.4 d since 2000. The average daily gain from weaning to sale during this same time increased from 585 to 795 g/d due to genetic advances and improved nutrition and housing, as well as sanitary conditions. Therefore, the risk of boar taint from entire males is no higher now than it was 20 years ago. The following paragraphs describe some practical considerations, both positive and negative, relating to nutrition when producing entire males.

Producing entire males also has some environmental benefits compared with surgical castrates. As previously pointed out, entire males fed the same diet as surgical castrates are more feed efficient, which means reduced manure production and ultimately lower nutrient losses from entire males. Lawlor et al. [86] found that N excretion per kg carcass was 11.7% less for entire males than for surgical castrates, principally due to increased N retention in entire males.

Entire males are much more aggressive than females and surgical castrates. The highest levels of aggression and sexual mounting are found in single-sex groups of entire males, with intermediate levels in mixed sex groups and the lowest levels in groups of females [87]. Therefore, entire male pigs appear to have better welfare when reared in mixed-sex groups, while females are better in single-sex groups, where they are not stressed by mounting and fighting caused by the males. Therefore, making recommendations on the optimal group gender composition is difficult. In the majority of cases in commercial pig production, entire males and females are not grouped separately and the same diet is fed to both sexes. In many cases, only one diet is fed to both genders between 30 kg and up to 120 kg live weight. Where farms do group entire males and females separately, they generally still feed the same diet to both groups.

For this reason, the only advantage in separating the sexes on these farms is that male pens can be emptied earlier, as the male pigs reach their target slaughter weight earlier than the females. However, the differential in growth and feed efficiency between entire males and females means that female pigs can be fed diets that are lower in energy by ~4% and lower in SID amino acids by ~7%. If implemented, this has potential to reduce feed cost by ~2.4 cents/kg dead weight or €2.02/gilt pig [88]. Penning entire males and females separately also allows entire males to be brought to a higher sale weight compared to females at a given slaughter age. Alternatively, if entire males and females are produced to the same target slaughter weight, it allows entire male pens to be emptied earlier [88]. Another advantage of penning entire males and females separately is that it eliminates the possibility of females being pregnant at slaughter, which may be a problem where pigs are slaughtered at particularly heavy weights and hence are older and more mature at slaughter.

Allen et al. [89] recommended that Irish pig producers house mixed-sex groups on fully slatted floors and feed them reduced crude protein diets in liquid form to help reduce the incidence of boar taint in the entire males. To reduce skatole development, a frequent recommendation is to withhold feed from entire males for 26 h prior to slaughter. In practice, this poses a welfare problem, as modern pigs have large appetites and hunger could cause escalation of mounting and aggression. Further investigation is required regarding the implications of feed withdrawal on welfare and skatole levels in entire male pigs and females in case of mixed-sex groups.

## 10. Conclusions

This review describes the various ways that can be used to tackle the different issues of rearing entire male pigs and immunocastrates through nutrition. The goal of these strategies is to optimize and fully exploit the growth potential and efficiency of these nutrient-efficient animals. At the same time, with the same diet, producers should be able to reach low levels of boar taint, sufficient intramuscular fat content, and a fat tissue suitable for meat processing or ripening. Do we have the solutions for all of this? We have the main puzzle pieces, but they need further exploitation. The reason for this is that the available results are not always consistent due to various factors, such as the breeds, genetic lines, rearing conditions, and feeding regimes. In the future, a more holistic approach is needed that considers the impact of the whole diet, with overall benefits on pig performance, health, pork quality, and profitability.

## Figures and Tables

**Figure 1 animals-10-01950-f001:**
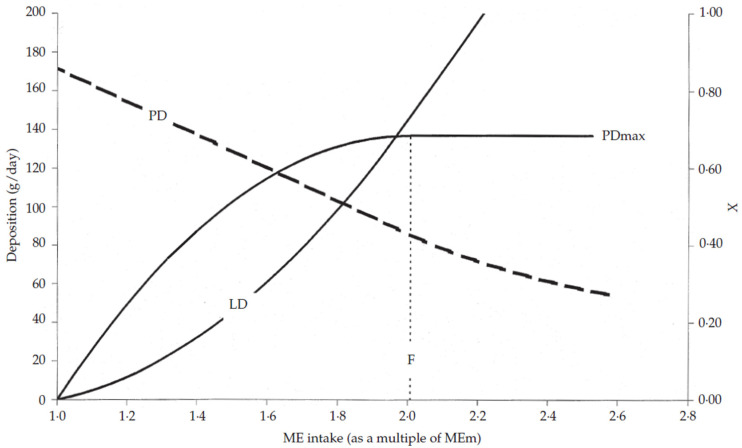
Model of the response of an individual animal to a changing energy supply. PD: protein deposition; LD: lipid deposition; PDmax: upper limit to protein deposition; F: level of energy intake (as a multiple of the maintenance energy requirement) to reach PDmax; X: fraction of metabolizable energy (ME) intake above maintenance designated toward PD (after Van Milgen et al. [12]).

**Figure 2 animals-10-01950-f002:**
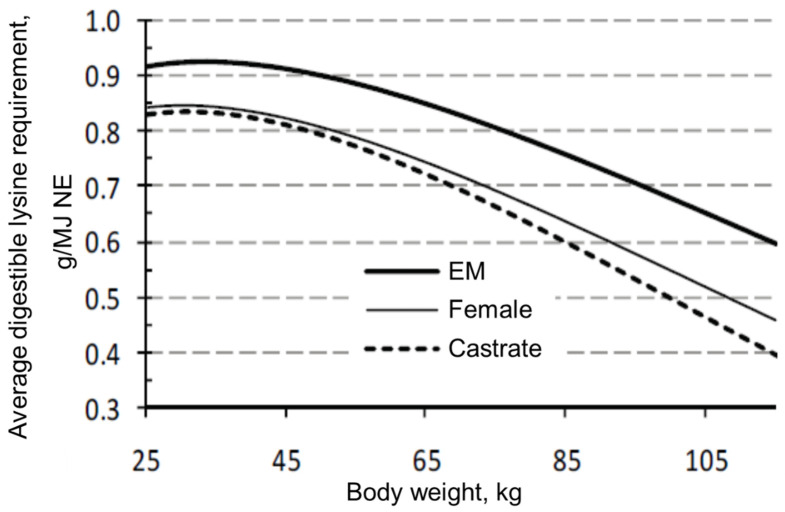
Estimation of digestible lysine requirements in pigs according to body weight and sex (after Quiniou et al. [26]).

**Figure 3 animals-10-01950-f003:**
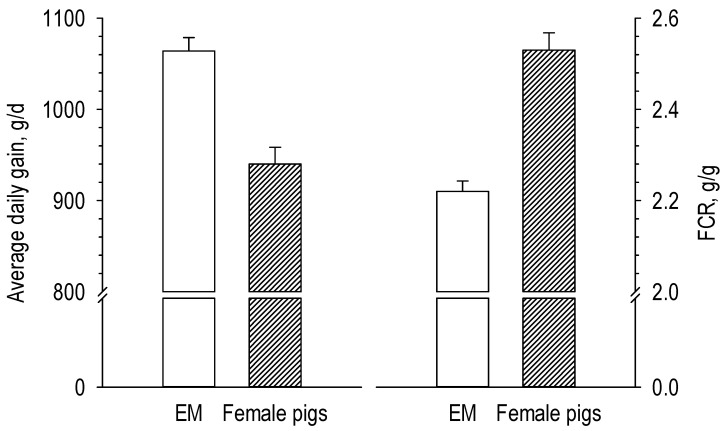
Average daily gain and feed conversion ratio of entire male (EM) and female pigs fed the same amount of lysine (0.5 g standard ileal digestible lysine/g digestible energy) (after Moore et al. [28]).

**Figure 4 animals-10-01950-f004:**
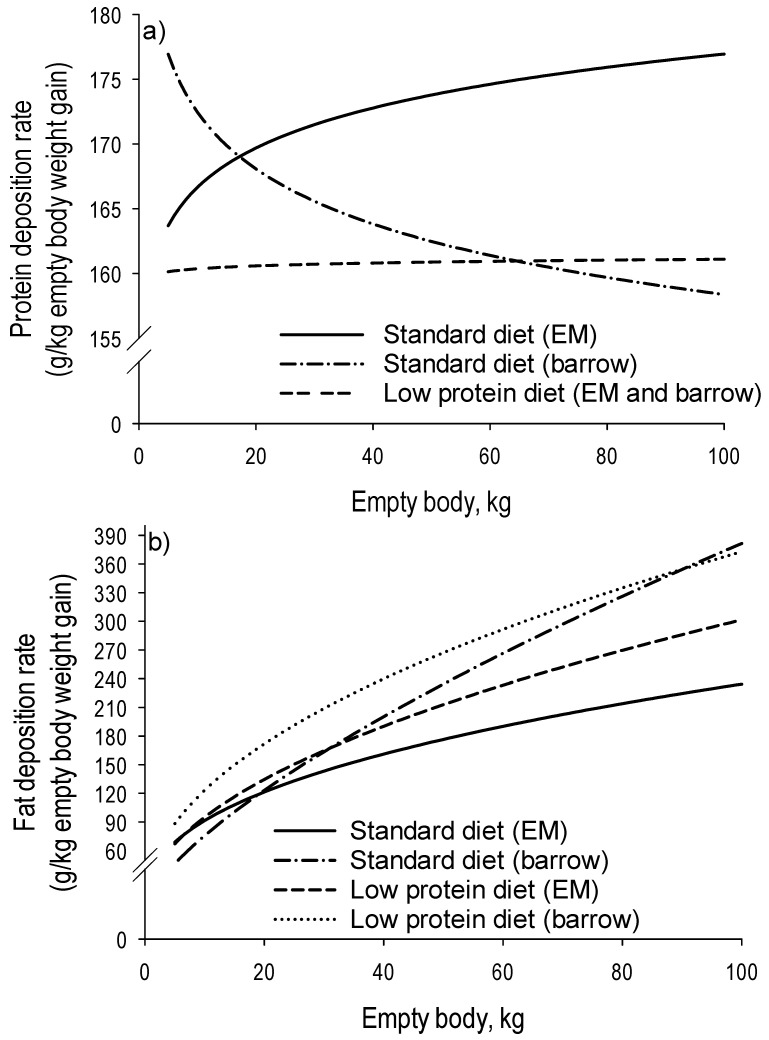
Dynamics of changes in relative deposition rates (g/kg empty body weight (BW) gain) of protein (**a**) and fat (**b**) with increasing empty BW of entire male pigs fed a standard or a low-protein, amino acid-limiting diet. The standard was formulated to meet nutrient requirements according to the standard Swiss feeding recommendations for grower-finisher pigs in the respective growth periods; the low protein diet was formulated to contain (expressed as percentage of the standard diet) 80% dietary crude protein, lysine, methionine + cystine, threonine and tryptophan (after Ruiz-Ascacibar et al. [2]).

**Figure 5 animals-10-01950-f005:**
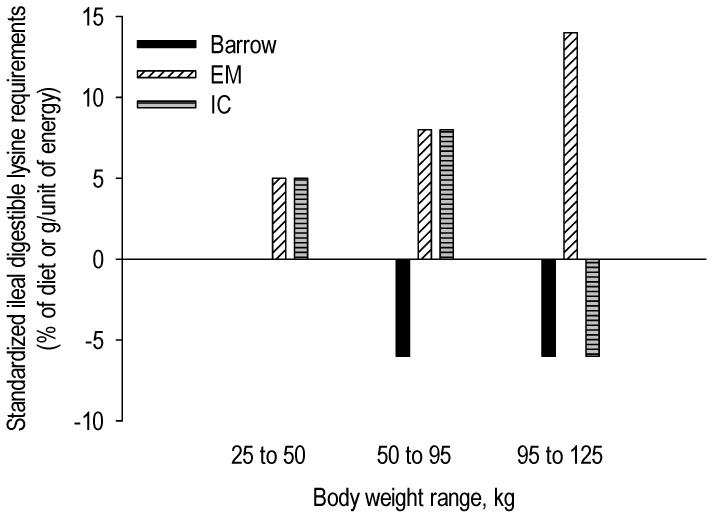
Suggested dietary standardized ileal-digestible lysine requirements in a three-phase feeding program adapted to meet requirements for female pigs, castrates, entire males (EM), and immunocastrates (IC). The standardized ileal-digestible lysine requirement levels are expressed relative to those for gilts (bases). The proposed schedule is to feed the diet of the third phase one week after the second immunization (after Dunshea et al. [13]).

**Table 1 animals-10-01950-t001:** Summary of some studies investigating the effects of dietary carbohydrates on skatole and indole levels in pigs.

Ingredient	Amount	Duration of Each Experimental Diet	Breed and Gender	Site of Measurement	Effect on Skatole	Effect on Indole	Reference
Chicory inulin	9%	From 67 kg body weight until slaughter at 101 kg	((Landrace × Yorkshire) × Landrace)) entire male pigs	Back fat	Decreased	No	Kjos et al. [40]
Chicory root	25%	7 days	(Duroc × (Danish Landrace × Yorkshire)) entire male pigs	Plasma	Tendency to be lower	No	Li et al. [41]
Oligofructose	5%	21 days	Danbred × Piétrain entire male pigs	Back fat	Decreased	No	Aluwé et al. [42]
Inulin	5%	21 days	Danbred × Piétrain entire male pigs	Back fat	Decreased	No	Aluwé et al. [42]
Short chain fructooligosaccharides	0.2%	from 60 kg body weight until slaughter at 130 kg (no information on exact duration)	White × Landrace cross entire male and female pigs	Back fat	Decreased	No	Salmon and Edwards [43]
Jerusalem artichoke	4.1, 8.1 and 12.2%	1 week	((Norwegian Landrace × Yorkshire) × (Norwegian Landrace × Duroc)) entire male pigs	Back fat	Tendency to be lower	No	Vhile et al. [44]
Hydrolysable tannin-rich chestnut wood extract	2% and 3%	From 123 to 190 days of age	Landrace × Large White entire male pigs	Back fat	Decreased compared to 1% supplementation	Not determined	Čandek-Potokar et al. [45]
Hydrolysable tannin-rich chestnut wood extract	1.5% and 3%	From 105 to 165 days of age	Swiss Large White entire male pigs	Back fat	Decreased compared to 1.5% but not 0% supplementation	Decreased	Bee et al. [46], Tretola et al. [47]
Raw potato starch	28%	From 120 to 170 days of age	Duroc × Landrace × Large White growing barrows	Colon	Decreased	Decreased	Zhou et al. [48]
Mixture of lignin and cellulose	8%	3 weeks	Duroc × Piétrain of both genders	Proximal and distal colon	Decreased	No	Pieper et al. [49]
Sugar beet pulp	12%	3 weeks	Duroc × Piétrain of both genders	Proximal and distal colon	No	Tendency to be lower	Pieper et al. [49]

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
