# Peer review of "Strategies to Meet Nutritional Requirements and Reduce Boar Taint in Meat from Entire Male Pigs and Immunocastrates"

_animals, 2020, doi:10.3390/ani10111950_

Round 1

Reviewer 1 Report

I consider the issue taken up in this article to be significant. The manuscript develops the subject logically and effectively, and scientific content of the paper justify its length. Paper is properly organized and clearly presented. Clearness and conciseness of manuscript’s writing style is good. Some minor comments (which in my opinion should be taken into account by the Authors) are included in the attached file.

L 64-66; is “The objective of this short review is to share new knowledge gathered during the COST Action entitled …”; correct “The objective of this short review is to share current knowledge gathered during the COST Action...”.  It is not "new" knowledge.

L 106-107;  Lack dot at the end of the sentence.

L 218-220; please provide a reference

L 220; “…because of its anti-inflammatory properties.” - please, give the appropriate references

L 274-275 ; please give the reference

L 306-307; these statement should be supported by literature data

References

  1. is “…de lange, C.F.M…”; correct “ …de Lange, C.F.M…”

Author Response

see attchement

Reviewer 2 Report

The paper provides an uptodate overview of knowledge on nutritional requirements for entire male pigs or immunocastrates. It also shares the example of Irish and UK pig production. The conclusion is sound.

Small remarks

L35: reformulate 'from purely from a point of view'

L178-179: why mentioning feed restriction as a good alternative just to contradict this statement in the following sentence ?

L184-186: does it mean that there is no good option for immunocastrates ?

L192: a reference for this statement ?

L204: Is there any explanation for the inconsistent results ?

L207-208: there is a layout problem

L209: what about the cost of inulin?

L216: it would be clearer to write that results on intestinal health were observed in other species.

L247-248: the sentence is confusing, does it mean that the differences were not significant?

L258: what about the cost of adding tannins in feed? or feasibility on a large scale? what would happen if females were fed with the same diet as entire males?

L296: Is there any study that combined several ingredient (inulin + tannins +...)?

L303: It means that meat from entire males has more water

L317: Is there any reason to believe that the relation between ingested and deposited PUFA would not be the same in entire males ? 

L318: Does'nt seem a realistic option

L367: repetition of L362

L370: What are the figures for occurence of pregnant females at slaughter house? If animals are slaughtered at a younger age, it should not be a problem.

L376: in case of mixed-sex groups, what about the welfare of females with feed withdrawal?
